# The Impact of Emotional Experience on Tourists' Cultural Identity and Behavior in the Cultural Heritage Tourism Context: An Empirical Study on Dunhuang Mogao Grottoes

**Yang Yang [1], Zhengyun Wang [2], Han Shen [1,2,*] and Naipeng Jiang [2]**

[1] School of History & Tourism, Hexi University, Zhangye 734000, China; 21110500004@m.fudan.edu.cn
[2] Tourism Department, Fudan University, Shanghai 200433, China; 13564109558@163.com (Z.W.); npjiang21@m.fudan.edu.cn (N.J.)
\* Correspondence: shen_han@fudan.edu.cn; Tel.: +86-021-55665028

**Abstract:** The emotions perceived by tourists and their effects in the tourism context are increasingly highlighted in tourism studies. In the cultural heritage tourism context, tourists' emotional experience comes from their cognitive evaluation of the natural environment and the humanistic environment and triggers deep cognitive processing and prosocial behavior, further building tourists' identity with culture and enhancing their awareness and heritage conservation behavior. Based on the theory of emotional evaluation and positive emotional expansion and construction, this study constructed the research model of emotional arousal—positive emotional experience—tourists' cultural identity—heritage protection behavior. Three hundred and ninety-seven tourists' data were empirically tested using the World Heritage Site, the Dunhuang Mogao Grottoes, as a case site. The study found that in the cultural heritage tourism context, the cognitive evaluation of the natural and humanistic environment has the effect of inducing positive emotional experience among tourists; positive emotional experience positively influences tourists' cultural identity and heritage conservation behavior; and they are part of the mediating variables of tourists' emotion elicitation and cultural identity. The results of this study will further enrich the theoretical research on emotions in the cultural heritage tourism context and also help the relevant departments of cultural heritage tourism further enhance tourists' cultural identity and heritage conservation behaviors from the perspective of tourists' emotional experience. The future research could focus on investigating the emotional triggers' impact on tourists' cultural identity and heritage conservation behavior in relation to a particular cultural experience activity.

**Keywords:** cultural heritage tourism; emotional experience; positive emotions; cultural identity; heritage conservation behavior

## 1. Introduction

In recent years, with the increasing experience of tourists and the rising awareness of tourists in pursuing a high-quality experience, the fundamental concern of the tourism industry practice has shifted from the straightforward "attractive tourist destinations" to the more intricate "unforgettable travel experience" [1]. The key to an unforgettable travel experience is the positive emotions and feelings associated with the experience [2]. Traveling can not only provide the experience of learning and cognitive dimension, but also stimulates a variety of emotional experiences such as pleasure, awe, and happiness [3]. Positive emotions have the effect of instantly awakening visitors' cognition and behavior, making them less self-focused psychologically and causing them to behave in a prosocial manner. In the cultural heritage tourism context, natural scenery, humanistic factors, activity experience, and service quality are objects of tourists' cognitive evaluation and are triggers that stimulate a wide range of emotions. Positive emotional experience not only increases tourists' satisfaction and loyalty to cultural heritage destinations [4] but also truly

satisfies tourists' inner feelings and identity needs, promoting tourists' identification of culture and heritage conservation behavior [5,6].

China is renowned for its lengthy history and rich culture. According to UNESCO [7], the total number of World Heritage Sites in China reached 56, including 38 World Heritage Sites, 4 World Cultural and Natural Heritage Sites, and 14 World Natural Heritage Sites, jumping to the first place in the world (tied with Italy). Chinese Cultural heritage is a comprehensive form of knowledge, technology, art, institutions, and beings formed by Chinese cultural history and is a crucial bearer of national identity and a crucial testament to the continuation [8]. Moreover, cultural heritage tourism is an important way to establish cultural identity [8]. Through the understanding of the social values and collective memory embedded in cultural heritage, it can not only enhance people's aesthetic ability [9], but also enhances cultural and national identity [10]. However, quantitative empirical studies on the identity of Chinese culture are still lacking [11]. Therefore, this study will focus on Chinese cultural heritage and explore the influence mechanism of cultural identity from the perspective of emotional experience.

Emotions are core to tourism experiences [12]. Research on tourist emotions at home and abroad has mainly focused on the triggers of tourism emotions and the effects of the emotional experience of tourism. Emotional evaluation theory [13] considered emotions the result of an individual's response to the cognitive evaluation of the surrounding environment or meaningful stimuli. Early scholars were keen to explore the dimensions and intensity of tourists' emotional experience, delineating the dimensions of tourists' emotional experience in different tourism contexts such as wildlife tourism [14], red tourism [15], black tourism [16], and mountain-based tourism [17]. In addition to the dimensions of emotional experience, it is more important to identify the stimuli in the tourism environment that stimulate tourists to trigger an emotional experience. Williams suggested that it was more meaningful to identify which factors in the tourism environment stimulate different positive emotional experiences for tourists and what impacts a positive emotional experience would have on tourists' behavior [18]. Tourists' personal perceptions and concerns in a tourism environment influence their perception of the surrounding ambiance, stimulating them to induce different emotional experiences by perceiving the intensity of stimuli of various environmental factors. Keltner's prototype theory of awe argued that awe stems from an individual's perceived evaluation of two aspects: one is the instantaneous shock and surprise of being confronted with powerful things and environments, and the other is the experience of psychological submission when exposed to cultural and spiritual aspects beyond one's perception [19]. The emotional experience of happiness positively influences revisit intention [20]. Xu et al. concluded through an empirical study that both the natural and humanistic environments in the tourism context can induce tourists' positive emotional experience, and in terms of the intensity of induced emotional experience, tourists perceive the humanistic environment factor stimuli more strongly than that of the natural environment. [21] In addition, factors such as innovativeness [22], the spatial scale of landscape [23], degree of space congestion [24], quality of tourist interaction [25,26], service facilities [27], challenge difficulty and individual skill matching degree [28], cultural attachment [29] and interaction between behavioral and atmospheric context [30] can all induce different emotional experiences. Most of the emotional triggers involve the natural environment, the humanistic environment, the cultural atmosphere, the interaction of tourism activity, and the individual characteristics of tourists.

Emotional experiences are not only about how tourists feel psychologically, but more importantly, how emotions tend to behave; for example, a negative emotional experience drives individuals to escape [31], while individuals with positive emotions are more willing to express themselves [32]. A growing body of research from both domestic and international scholars has demonstrated that the evaluation of tourists' emotional experience is an important factor influencing their behavioral tendencies, satisfaction, and loyalty. In the cultural tourism context, both perceived quality and emotion positively influence tourist satisfaction, and the emotional components play a moderating role in the influence of the

cognitive component on satisfaction [33]. Tourist satisfaction and loyalty are influenced by a combination of cognitive and emotional factors [34]. The study also provided further evidence of the importance of the positive emotions of tourists in influencing deeper cognitive experience. Positive affective awe has a positive effect on tourists' environmentally responsible behavior, national identity behavior, and tourists' pro-environmental behavior [35].

In summary, most existing research studies have explored the emotional experience in a particular tourism context, and most of the studies on the tourists' emotional experience focused on positive emotions. Only a small number of studies in China have mentioned the antecedent variables of emotion generation, and most of them are studies of "awe-inspiring emotions" [36]. Quantitative empirical studies on Chinese culture identity from the perspective of emotional experience are still lacking [11]. This paper conducts a pathway model of emotional triggers, emotional experience, and emotional influences in the cultural heritage tourism context. The aims of this paper are as follows: (1) exploring the dimensions and quality of tourists' emotional experience in the cultural heritage destination and (2) exploring whether the natural environment and humanistic environment of cultural heritage destination are the factors that induce tourists' emotional experience, as well as the degree of influence of different factors on different dimensions of emotional induction and (3) exploring whether the natural environment and humanistic environment in the cultural heritage affect cultural identity and heritage protection behavior with the medium of tourists' emotional experience.

This study makes two contributions. At the theoretical level, this study will further expand and deepen the theoretical research on emotional experience in the cultural heritage tourism context by effectively revealing the influence mechanisms of tourists' emotional experience on cultural identity and heritage conservation behavior. At the practice level, this study offers empirical research references for tourism businesses and cultural heritage scenic areas that will help them create cultural products from the perspective of visitors' emotional experiences in order to fully satisfy visitors' demand for cultural tourism experiences, foster visitors' sense of cultural identity, and raise visitors' awareness of the need to protect cultural heritage.

## 2. Theoretical Basis and Research Hypothesis

### 2.1. Theory of Emotional–Cognitive Evaluation

In the 1960s, Magda Arnold put forward the theory of emotion evaluation and defined the generation conditions of emotions. Emotions arise from the individual's cognitive evaluation of stimulus in the environment, and the stimulus in the environment does not directly determine the production of emotions. Individuals will experience different emotions when they are in the same environment and have different "cognition—evaluation" processes in response to the same stimulus [37]. Arnold further suggested that when a stimulus in the environment is cognitively evaluated by an individual and triggers an emotional experience, cognition and emotion will act together in the generation of motivation and behavior, which will then manifest in individual behavioral trends.

Richard Lazarus [38] inherited and developed Arnold's theory of emotional evaluation and proposed the theory of emotional derivative theory—cognitive evaluation theory. He argued that the environmental stimuli on the individual's interests determine the type of emotion he feels, and that each specific emotion is derived from a combination of three characteristics: cognitive evaluation, activity tendencies, and physiological changes [38]. Lazarus particularly emphasized that emotion is a response to meaning, which is accomplished by the individual's cognitive evaluation process. The fact that the same event produces different emotions for different people stems from the fact that environmental events have different meanings for different people. Lazarus' theory of cognitive evaluation also pointed to the mechanism by which individual behavior is generated, whereby emotions lead individuals to cope with behavioral tendencies when the relational meaning of an individual's cognitive evaluation affects individual desire or motivation. In other words, in-

dividuals need to complete the continuous process of the "cognitive evaluation—emotional experience—coping behavior" continuum [39].

Moors argued that the cognitive evaluation theory laid down by Lazarus, which explained the link between cognitive evaluation and emotion generation, has become a core theory in the field of emotion research [40]. Findings related to tourists' emotional experience suggest that the research focus on tourists' emotional experience has shifted from explaining how emotions influence people's motivation and behavioral responses to a more in-depth exploration of which stimuli in a situation can induce different emotional experience in individuals. Emotional evaluation theory and its derivative, cognitive evaluation theory, provide the theoretical foundation and support for research on emotions in the field of tourism.

### 2.2. Broaden-and-Build Theory of Positive Emotions

In 2001, Fredrickson, an American psychologist, developed the broaden-and-build theory of positive emotions based on emotional evaluation theory [41]. Fredrickson defined the unique function of positive emotions as extended and constructive. Any positive emotional experience perceived by an individual has the potential to expand instantaneous thinking, which in turn builds the individual's resources such as intellectual, physical and psychological, and social resources, thus bringing long-term positive effects to the individual [42]. The expansive effects of positive emotions are manifested in the ability to instantly increase an individual's thinking—range of action, which allows for the sustainable building of personal resources and ultimately brings positive benefits to the individual. By perceiving the benefit effect, the individual will further induce positive emotional experiences, positive emotions, and individual growth, forming a spiraling cycle [32]. Fredrickson was the first to propose four key positive emotions, namely happiness, interest, satisfaction, and love, and later he added a total of ten positive emotions such as pride, admiration hope, etc. He also emphasized that there are more than ten positive emotions mentioned above that can enhance individual expansion and construction [31].

Numerous studies have shown that positive emotions allow individuals to become creative, integrated, forward looking, and high level in their thinking patterns [43–45]. Positive emotions can promote intellectual resources such as the ability to learn and problem-solving skills with a more open perspective. They can lead individuals to build physical resources, such as physical strength and physical health, more actively; improve psychological resources such as resilience to overcome difficulties and a sense of identity; and provide social resources by helping build new social connections [32]. Positive emotions lead to positive outcomes in thinking and resources, and positive outcomes continue to induce a positive emotional experience, creating a good cycle of interaction between positive emotions and positive outcomes. Individuals in a positive emotional state are able to view events more open mindedly and are more likely to perceive the positive implications of adversity, which brings long-term benefits to the individual. These in turn continue to trigger positive emotional experiences [46].

### 2.3. Research Hypothesis and Theoretical Model

Emotional evaluation theory suggests that the process of cognitive evaluation of environmental stimulus is a determinant of the generation of emotions. Tourists' emotional experiences arise from the process of cognitive evaluation of stimuli in the tourism environment, such as the natural and humanistic environment. In a study of souvenir design based on emotional memory, Zhu et al. [47] found that humanistic factors of the destination, such as tourist souvenirs, aroused tourists' positive emotional experience. Both natural and humanistic environmental factors can stimulate tourists' positive emotions, and heritage revitalization experience with rich cultural factors such as traditional folklore, song and dance, and other dynamic experiential interactions can induce a greater stimulating effect than the natural environment [21]. From a tourist perception perspective, Yao et al. [48] found that children have a greater preference for experiential activities that require hands-

on experience, and this participation can enrich the tourism experience for children and parents. Hou [49] found that cultural experience activity in digitalized museums received a high degree of attention from tourists for their novel visual experience and spiritual enjoyment, which improved the quality of their tourism experience. The Dunhuang Mogao Grottoes were used as a case study site and the humanistic environment of the Dunhuang Mogao Grottoes included entity caves and cultural experience activities [50]. Accordingly, this study proposes the following research hypotheses:

**Hypothesis 1.** *The cognitive evaluation of the natural environment positively influences the tourist's emotional experience.*

**Hypothesis 2.** *The cognitive evaluation of entity caves positively influences the tourist's emotional experience.*

**Hypothesis 3.** *The cognitive evaluation of the cultural activity experience positively influences the tourist's emotional experience.*

The theory of positive emotion expansion and construction states that positive emotional experience can instantaneously expand one's attention and cognitive scope, recognize the positive aspects of the event with a more open mind, and construct individual psychological resources such as identity. Wu et al. [51] took the intangible cultural heritage of local opera as the research object and found that emotional experience based on local opera has a positive impact on cultural identity. Liu [15] found that visitors evoke individual emotions and ultimately act on their own changes in cognitive attitudes towards individuals by engaging in red tourism activity experience, such as cultural cognition and national identity. Through an empirical study, Liu et al. [52] showed that tourists with significant emotional identity are more likely to show heritage conservation behavioral intentions and thus implement heritage conservation behaviors. Cheng & Chen [29] showed that tourists' local attachment to cultural heritage scenic spots has an impact on their responsible environmental behaviors. Accordingly, the following hypotheses are formulated:

**Hypothesis 4.** *Tourists' emotional experiences have a positive impact on cultural identity.*

**Hypothesis 5.** *Tourists' emotional experience positively influences heritage conservation behavior.*

Song [53] conducted a study on cultural heritage conservation and cultural identity in the process of urbanization and verified that cultural identity and cultural heritage conservation are confirmed to be intrinsically related. Cultural identity is the core mechanism for the inheritance of intangible cultural heritage [54]. Wang et al. [55] concluded from an empirical study of residents of the Huizhou Cultural and Ecological Reserve that the cultural identity of local residents has a significant positive influence on cultural conservation behavior. Accordingly, the following hypothesis is proposed:

**Hypothesis 6.** *Cultural identity has a positive impact on heritage conservation behavior.*

During cultural heritage tourism activities, when a tourist gazes at cultural landscapes and sites, a sense of cultural identity and belonging are inspired. The richness of natural and humanistic resources in cultural heritage sites and the interesting interpretation are antecedent variables that stimulate tourists' cultural self-confidence and emotional identity [56]. Bowen & Giannini et al. [57] argued that tourists' participation in digital experience activities in museums is more likely to evoke personal memories, and emotional experience contributes to enhancing tourists' cultural identity compared to traditional forms. Whether tourists know each other or not, the intangible atmosphere of cultural heritage evokes their collective memories and shared emotions [58]. Based on this, the study proposes Hypothesis 7:

**Hypothesis 7.** *The cognitive evaluation of the humanistic environment positively influences tourists' cultural identity.*

Some scholars have found through empirical research that in the red cultural heritage tourism context, there is a mediating effect of awe in positive emotions between tourists' awe-inducing emotions and national identity [59]. Similarly, in the cultural heritage tourism context, there may be a mediating effect of positive emotions between the cognitive evaluation of the natural environment and cultural identity and between the cognitive evaluation of the humanistic environment and cultural identity. Based on this, the study proposes Hypothesis 8 and Hypothesis 9.

**Hypothesis 8.** *Positive emotions moderate the impact of cognitive evaluation of the natural environment on cultural identity.*

**Hypothesis 9.** *Positive emotions moderate the impact of the cognitive evaluation of the humanistic environment on cultural identity.*

Based on emotional evaluation theory, positive emotion expansion theory, and the existing empirical studies mentioned above, the hypothesis of this study is proposed and the hypothesis model is shown in Figure 1.

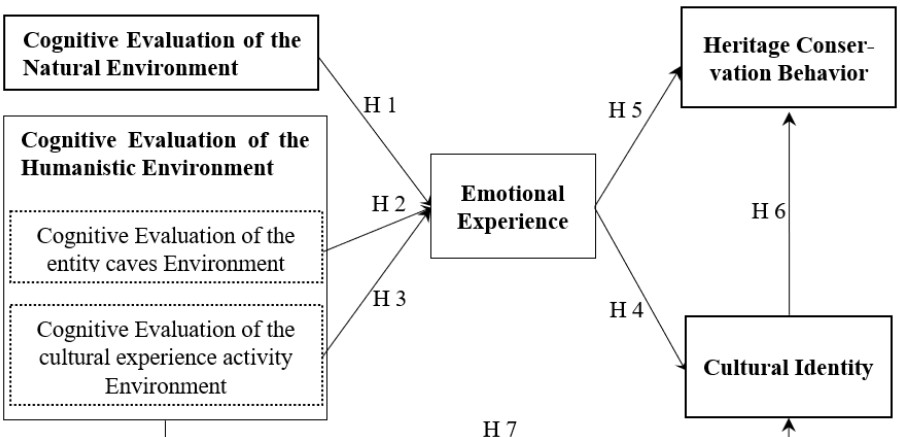

**Figure 1.** Theoretical model.

## 3. Research Design

### 3.1. Introduction of the Case Site

This study takes the world cultural heritage site—Dunhuang Mogao Grottoes scenic area—as a case study object. The Mogao Grottoes, located on the cliff of the south-eastern corner of Dunhuang, the historical and cultural city and a strategic passage on the ancient Silk Road, have been an art treasure house for nearly 1700 years and were listed as a World Cultural Heritage site in 1987.

With its long history and abundant natural landscape, the Mogao Grottoes are the shining jewel on the ancient Silk Road. To meet visitors' needs for a deeper experience of Dunhuang culture, the Dunhuang Academy has developed a series of scenario-based and immersive cultural experiences, such as the sight-seeing experience of digital Mogao Grottoes on a ball screen, animation production of Dunhuang murals, a mural copying in ancient style, mural painting restoration, and ancient costume and etiquette experience of mural painting, etc. The long history and culture, rich natural landscape and differentiated high-quality cultural activity contribute to a positive emotional experience for visitors to Mogao Grottoes and maximize their demand for different depths of the Dunhuang cultural experience.

### 3.2. Measurement Scales and Questionnaire

The variables and indicators of measurement are selected from existing established scales and adapted to our research topic. The scale was discussed with several experts in the field of tourism and the managers of the Mogao Grottoes Scenic Area, and the content of the questionnaire items was fine-tuned to ensure the scientific validity and rigor of the measurement instrument. The initial questionnaire was pre-surveyed by 50 visitors who had experienced the Mogao Grottoes. The results of the pre-study showed that the questionnaire had good reliability and validity. No further other corrections to the questionnaire were needed, so it could be released directly as a formal questionnaire.

The selected scales are as follows: the natural environment cognitive evaluation scale was measured with four items adapted from Tang et al. [60] and applied to the Nianhua Bay of Lingshan Humanistic Town [21] and the World Natural Heritage Wulong Scenic Spot [61], all of which have achieved satisfactory results. The evaluation of the entity caves of the Mogao Grottoes in the cognitive evaluation of the humanistic environment was measured with four items based on the work of Tang et al. [60] and Gao et al. [62]. The cognitive evaluation of the humanistic environment was measured with six items based on Liu [63] for the quality of tourism performance products, Zhan et al. [64] for the evaluation of experiential tourism activities in agricultural cultural heritage, and Qin [65] for the dimensional evaluation of the ethnic festival experience. The tourist emotional experience scale was measured using a 12-item scale based on the Destination Emotion Scale (DES) which was constructed by Hosany and Gilbert [66] and developed by Sun et al. [67]. The positive negative emotion scale is based on the scale used by Liu & Yue [15] to measure positive negative emotions in red tourism situations, with three items. The cultural identity scale is based on He & Ai's study [56] on the influence of national archaeological park tourism experience on tourists' cultural identity, with four items. The heritage conservation behavior scale draws on the scale on heritage conservation behavior in the cultural heritage of the Jiayuguan Guancheng scenic area, with four items [52]. For the English scales involved in the study, a standardized "double translation" process was followed to ensure the accuracy of the translation. Based on the measurement scales, we further subdivide the hypotheses into 25, as shown in Table 1.

### 3.3. Sample and Data Collection

The study used the convenient sampling method and distributed questionnaires online through channels such as WeChat friend circle and online community. To ensure the quality, the respondents were informed about the purpose of the study and ensured anonymity. The respondents answered a filtering question "Have you ever visited Dunhuang Mogao Grottoes scenic area?". Only those who answered "Yes" were invited to continue the questionnaire. The official questionnaire was distributed via the internet for data collection in September 2021. The total number of questionnaires distributed was 450, all of which were collected. Fifty-three questionnaires with incomplete responses and all responses being the same were excluded. A total of 397 valid questionnaires was obtained, with an efficiency rate of 88.2%. As shown in Table 2, 60.7% respondents were female. The percentage of males was 39.29%. Of the respondents, 10.86% were under 20 years of age, 2.24% were between 21 and 30 years old, 43.58% were between 31 and 40 years old, 6.63% were between 41 and 50 years old, 3.78% were between 51 and 60 years old, and 2.91% were over 60 years old. In terms of education, 85.89% were college/bachelor's degree holders, 9.07% were master's degree holders and for average monthly income, 46.60% of visitors were earning between RMB 5000 and 10,000 and 31.74% between RMB 10,001 and 20,000. The majority of visitors had a higher income level.

**Table 1.** Hypothesis in the study.

| Hypothesis | Contents |
|---|---|
| Hypothesis 1 | H1a Cognitive evaluation of the natural environment positively influences visitors' emotion of surprise<br>H1b Cognitive evaluation of the natural environment positively influences visitors' emotion of delight<br>H1c Cognitive evaluation of the natural environment positively influences visitors' emotion of happiness<br>H1d Cognitive evaluation of the natural environment positively influences visitors' negative emotions |
| Hypothesis 2 | H2a Cognitive evaluation of the entity caves positively influences visitors' emotion of surprise<br>H2b Cognitive evaluation of the entity caves positively influences visitors' emotion of delight<br>H2c Cognitive evaluation of the entity caves positively influences visitors' emotion of happiness<br>H2d Cognitive evaluation of the entity caves positively influences visitors' emotion of negative emotions |
| Hypothesis 3 | H3a Cognitive evaluation of the cultural activity experience positively influences visitors' emotion of surprise<br>H3b Cognitive evaluation of the cultural activity experience positively influences visitors' emotion of delight<br>H3c Cognitive evaluation of the cultural activity experience positively influences visitors' emotion of happiness<br>H3d Cognitive evaluation of the cultural activity experience positively influences visitors' negative emotions |
| Hypothesis 4 | H4a Visitors' amazement positively influences their cultural identity<br>H4b Visitors' delight positively influences their cultural identity<br>H4c Visitors' happiness positively influences their cultural identity<br>H4d Visitors' negative emotions positively influence their cultural identity |
| Hypothesis 5 | H5a Visitors' amazement positively influences their heritage conservation behavior<br>H5b Visitors' delight positively influences their heritage conservation behavior<br>H5c Visitors' happiness positively influences their heritage conservation behavior<br>H5d Visitors' positive negative emotions influence their heritage conservation behavior |
| Hypothesis 6 | H6 Visitors' cultural identity positively influences their heritage conservation behavior |
| Hypothesis 7 | H7a Cognitive evaluation of entity caves positively influences visitors' cultural identity<br>H7b Cognitive evaluation of cultural activity experience positively influences visitors' cultural identity |
| Hypothesis 8 | The positive emotions moderate the impact of cognitive evaluation of the natural environment on cultural identity |
| Hypothesis 9 | The positive emotions moderate the impact of the cognitive evaluation of the humanistic environment on cultural identity |

**Table 2.** Respondent's profile ($n$ = 397).

| Title | Option | Frequency | Percentage (%) |
|---|---|---|---|
| Gender | Male | 156 | 39.29 |
| | Female | 241 | 60.71 |
| Age | 18~25 | 59 | 14.86 |
| | 26~30 | 128 | 32.24 |
| | 31~40 | 173 | 43.58 |
| | 41~50 | 22 | 5.54 |
| | 51~60 | 15 | 3.78 |
| Education | Junior high school and below | 1 | 0.25 |
| | High school | 19 | 4.79 |
| | College/bachelor's degree | 341 | 85.89 |
| | Master's degree and upper | 36 | 9.07 |
| Monthly income | Under 5000 | 39 | 9.82 |
| | 5000~10,000 RMB | 185 | 46.6 |
| | 10,001~20,000 RMB | 126 | 31.74 |
| | 20,001~30,000 RMB | 28 | 7.05 |
| | Over 30,000 RMB | 2 | 0.5 |

## 4. Data Analysis

### 4.1. Reliability Detection

The reliability and validity of the measurement model can be tested by the standardized factor loadings of the items, Cronbach's α, composite reliability (CR), and average variance extracted (AVE). The best reliability combinations were found when the standardized factor loadings were higher than 0.6, Cronbach's α was greater than 0.7, CR was greater than 0.7, and AVE was greater than 0.5, indicating ideal reliability of the observed variables on the latent variable measures [68]. In the study, the reliability of the questionnaire was first tested to ensure the reliability of the scale. As shown in Table 3, Cronbach's α values of each latent variable in this paper range from 0.857 to 0.936, all of which are greater than 0.8, indicating that the scale has good internal consistency and high reliability.

**Table 3.** Results of reliability and convergent validity tests (*n* = 397).

| Variables and Measurement Items | Factor Loading | Cronbach's α | AVE | CR |
|---|---|---|---|---|
| Cognitive Evaluation of the Natural Environment (CENE) | | 0.808 | 0.512 | 0.807 |
| The natural landscape of the Mogao Grottoes is well preserved. | 0.771 | | | |
| The natural beauty of the Mogao Grottoes is delightful. | 0.714 | | | |
| The natural landscape of the Mogao Grottoes is integrated into the caves. | 0.712 | | | |
| Fulfilling my imagination of the natural landscape of the North West. | 0.671 | | | |
| Cognitive Evaluation of the Entity Caves (CEEC) | | 0.857 | 0.609 | 0.86 |
| The sculptures in the Mogao Grottoes are superbly carved. | 0.78 | | | |
| The murals in the caves are absolutely beautiful. | 0.818 | | | |
| The docents of Maogao Grottoes are very professional. | 0.668 | | | |
| A visit to the Mogao Grottoes gave me a taste of the world's cultural heritage | 0.84 | | | |
| Cognitive Evaluation of the Cultural Activity Experience (CECAE) | | 0.857 | 0.545 | 0.857 |
| The cultural experience activity has increased my knowledge of traditional Chinese culture. | 0.767 | | | |
| The presentation of cultural experience has shown the authenticity of cultural heritage. | 0.74 | | | |
| The cultural experience activity is highly innovative | 0.683 | | | |
| The scenic setting of the cultural experience activity is keeping the theme. | 0.759 | | | |
| The interpreters of the cultural experience activity are professional. | 0.745 | | | |
| Surprise (S) | | 0.892 | 0.741 | 0.895 |
| I feel surprised. | 0.84 | | | |
| I am shocked. | 0.897 | | | |
| I am in awe. | 0.84 | | | |
| Pleasure (P) | | 0.876 | 0.702 | 0.876 |
| I feel relaxed. | 0.832 | | | |
| I am comfortable. | 0.801 | | | |
| I am delighted. | 0.875 | | | |
| Happiness (H) | | 0.866 | 0.689 | 0.869 |
| I am happy. | 0.873 | | | |
| I am moved. | 0.824 | | | |
| I feel an attachment. | 0.794 | | | |
| Positive Negative Emotions (PNE) | | 0.935 | 0.835 | 0.938 |
| I feel sad. | 0.938 | | | |
| I feel sorry. | 0.861 | | | |
| I feel frustrated. | 0.932 | | | |
| Cultural Identity (CI) | | 0.87 | 0.629 | 0.871 |
| Learn more about Dunhuang culture than before. | 0.763 | | | |
| During the tour, I developed a strong sense of national pride and cultural self-confidence. | 0.81 | | | |
| I would have spent more time exploring the Mogao Grottoes if I could have. | 0.776 | | | |
| I would like to introduce the culture of Dunhuang I learned about today to my friends, family, and classmates around me. | 0.816 | | | |
| Heritage Conservation Behavior (HCB) | | 0.931 | 0.773 | 0.932 |
| I will take legal action to stop the destruction of my heritage. | 0.863 | | | |
| I abide by the rules and regulations of the cultural heritage site. | 0.881 | | | |
| I would strongly discourage the acts that harm cultural heritage if I see them. | 0.866 | | | |
| I took great care of the buildings and facilities when I visited and browsed. | 0.908 | | | |

### 4.2. Validity Detection

In the study, the confirmatory factor analysis was first conducted to test for convergent validity. The results of the convergent validity tests are shown in Table 3. The normalized

factor loading for all the measured items is greater than 0.6, the AVE values are greater than 0.5 and the CR values for each variable are greater than 0.7, indicating that all variables have high convergent validity [68]. In addition, all variables were tested for differential validity in this paper. As shown in Table 4, the square root of the AVE of each variable on the diagonal is greater than the correlation coefficient between the corresponding variables, indicating that the variables have good discriminant validity [68].

**Table 4.** Results of the discriminant validity test.

|  | CENE | CEEC | CECAE | S | P | H | PNE | C I | HCB |
|---|---|---|---|---|---|---|---|---|---|
| CENE | 0.716 | | | | | | | | |
| CEEC | 0.381 | 0.78 | | | | | | | |
| CECAE | 0.233 | 0.334 | 0.739 | | | | | | |
| S | 0.334 | 0.44 | 0.34 | 0.861 | | | | | |
| P | 0.357 | 0.418 | 0.253 | 0.582 | 0.838 | | | | |
| H | 0.293 | 0.364 | 0.297 | 0.578 | 0.546 | 0.83 | | | |
| PNE | 0.107 | 0.187 | 0.112 | 0.26 | 0.278 | 0.269 | 0.914 | | |
| CI | 0.321 | 0.478 | 0.411 | 0.534 | 0.512 | 0.558 | 0.304 | 0.793 | |
| HCB | 0.273 | 0.414 | 0.239 | 0.615 | 0.572 | 0.619 | 0.404 | 0.57 | 0.879 |

*4.3. Main Effect Test*

In this paper, the model was tested for fit by selecting the aptitude indicators such as CMIN, DF, GFI, CFI, IFI, TLI and so on [69], with Amos 24.0. The results are shown in Table 5. The CMIN value is 1174.635, the fit indicator CMIN/DF is less than 3, the GFI-CFI fit indicators are all greater than 0.8 and the RMSEA is less than 0.08, meeting the acceptable fit criteria for the model [69]. The model fit results indicate that no correction to the model is required and that the overall fit of the model is acceptable and can be further tested for the main effects. The specific results of the model tests are shown in Table 5. The results of the main effect analysis are shown in Table 6.

The path coefficients of natural environment cognitive evaluation on surprise, pleasure, and happiness are all greater than 0, and the *p*-value is less than 0.05, indicating that all three path coefficients are significant, so the three hypotheses H1a, H1b, and H1c are all supported. The path coefficient of natural environment evaluation on positive negative emotion is greater than 0, but the *p*-value is 0.238 (*p* > 0.05), indicating that this path coefficient is significant, so hypothesis H1d is not supported.

The path coefficients for the cave entity cognitive evaluations of surprise, pleasure, happiness, and positive negative emotion are all greater than 0, and the *p*-values are less than 0.05, indicating that all four path coefficients are significant, so H2a, H2b, H2c, and H2d are supported.

The path coefficients of the cognitive evaluation of cultural activity experience for surprise, pleasure, and happiness are all greater than 0, and the *p*-value is less than 0.05, indicating that all three path coefficients are significant, so all three hypotheses, H3a, H3b, and H3c, are supported. The path coefficient of the evaluation of cultural activity experience for positive negative emotion is greater than 0, but the *p*-value is 0.217 (*p* > 0.05), indicating that this path coefficient is not significant, so hypothesis H3d is not supported.

The path coefficients of surprise, pleasure, happiness, and positive negative emotion on cultural identity are all greater than 0, and the *p*-value is less than 0.05, indicating that all four path coefficients are significant, so the hypotheses H4a, H4b, H4c, and H4d are all supported. The path coefficients of surprise, pleasure, happiness and positive negative emotion on heritage conservation behavior are all greater than 0, and the *p*-value is less than 0.05, indicating that all four path coefficients are significant, so hypotheses H5a, H5b, H5c, and H5d are all supported.

The path coefficient of visitors' cultural identity on heritage conservation behavior is greater than 0 and the *p*-value is less than 0.05, indicating that this path coefficient is significant, so the hypothesis of H6 is supported. The path coefficients of cave entity

cognitive evaluation and cultural activity experience cognitive evaluation on visitors' cultural identity are greater than 0 and the *p*-value is less than 0.05, and H7a and H7b are supported.

**Table 5.** Model fitting result.

| Model Indexes | CMIN | DF | CMIN/DF | GFI | NFI | RFI | IFI | TLI | CFI | RMSEA |
|---|---|---|---|---|---|---|---|---|---|---|
| Value | 1174.635 | 472 | 2.489 | 0.840 | 0.871 | 0.856 | 0.919 | 0.908 | 0.918 | 0.061 |
| Suggested Values | | | <3 | >0.8 | >0.8 | >0.8 | >0.8 | >0.8 | >0.8 | <0.08 |
| Judgment of Results | - | - | achieved | achieved | achieved | achieved | achieved | achieved | achieved | achieved |

**Table 6.** Results of main effect analysis.

| Observational Variable | | Latent Variable | Estimate | S.E. | C.R. | *p* | R² |
|---|---|---|---|---|---|---|---|
| S | ← | ENE | 0.375 | 0.08 | 4.661 | *** | 0.065 |
| P | ← | ENE | 0.488 | 0.087 | 5.626 | *** | 0.103 |
| H | ← | ENE | 0.446 | 0.101 | 4.435 | *** | 0.063 |
| PNE | ← | ENE | 0.175 | 0.149 | 1.179 | 0.238 | 0.004 |
| S | ← | ENC | 0.481 | 0.065 | 7.423 | *** | 0.167 |
| P | ← | ENC | 0.47 | 0.067 | 6.967 | *** | 0.15 |
| H | ← | ENC | 0.469 | 0.078 | 5.99 | *** | 0.11 |
| PNE | ← | ENC | 0.43 | 0.116 | 3.705 | *** | 0.041 |
| S | ← | ECAE | 0.316 | 0.068 | 4.642 | *** | 0.06 |
| P | ← | ECAE | 0.185 | 0.07 | 2.629 | 0.009 | 0.019 |
| H | ← | ECAE | 0.327 | 0.085 | 3.867 | *** | 0.044 |
| PNE | ← | ECAE | 0.157 | 0.127 | 1.235 | 0.217 | 0.005 |
| CI | ← | S | 0.12 | 0.037 | 3.217 | 0.001 | 0.03 |
| CI | ← | S | 0.1 | 0.039 | 2.538 | 0.011 | 0.019 |
| CI | ← | H | 0.171 | 0.032 | 5.269 | *** | 0.082 |
| CI | ← | PNE | 0.049 | 0.018 | 2.651 | 0.008 | 0.015 |
| CI | ← | EEC | 0.186 | 0.051 | 3.631 | *** | 0.048 |
| CI | ← | ECAE | 0.21 | 0.049 | 4.298 | *** | 0.051 |
| HCB | ← | P | 0.211 | 0.056 | 3.759 | *** | 0.034 |
| HCB | ← | S | 0.32 | 0.058 | 5.473 | *** | 0.073 |
| HCB | ← | H | 0.292 | 0.052 | 5.603 | *** | 0.088 |
| HCB | ← | PNE | 0.143 | 0.028 | 5.042 | *** | 0.047 |
| HCB | ← | CI | 0.243 | 0.101 | 2.414 | 0.016 | 0.022 |

Notes: *** *p* < 0.001.

*4.4. Mediation Effect Test*

A mediation model is needed to test whether there is a significant or partial mediation effect. Regression analyses were conducted using SPSS software to verify the mediating effect of positive emotions in the cognitive evaluation of the natural environment, the humanistic environment, and cultural identity.

As can be seen from Table 7 below: the test of the mediating effects involves a total of three models, which are as follows:

Cultural Identity (CI) = 3.640 + 0.383 × Cognitive Evaluation of the Natural Environment (CENE)

Positive Emotions (PE) = 2.305 + 0.519 × Cognitive Evaluation of the Natural Environment (CENE)

Cultural Identity (CI) = 2.410 + 0.106 × Cognitive Evaluation of the Natural Environment (CENE) + 0.534 × Positive Emotions (PE)

**Table 7.** A test of the mediating effect of PE between CENE and CI.

| | CI | | | | | PE | | | | | CI | | | | |
|---|---|---|---|---|---|---|---|---|---|---|---|---|---|---|---|
| | B | Standard Error | t | p | β | B | Standard Error | t | p | β | B | Standard Error | t | p | β |
| Constants | 3.640 ** | 0.338 | 10.776 | 0 | - | 2.305 ** | 0.37 | 6.229 | 0 | - | 2.410 ** | 0.288 | 8.381 | 0 | - |
| CENE | 0.383 ** | 0.057 | 6.732 | 0 | 0.321 | 0.519 ** | 0.062 | 8.339 | 0 | 0.387 | 0.106 * | 0.05 | 2.108 | 0.036 | 0.089 |
| PE | | | | | | | | | | | 0.534 ** | 0.037 | 14.301 | 0 | 0.6 |
| $R^2$ | | 0.103 | | | | | 0.15 | | | | | 0.409 | | | |
| Adjusted $R^2$ | | 0.101 | | | | | 0.148 | | | | | 0.406 | | | |
| F value | | $F_{(1, 395)} = 45.320, p = 0.000$ | | | | | $F_{(1, 395)} = 69.539, p = 0.000$ | | | | | $F_{(2, 395)} = 136.592, p = 0.000$ | | | |

$* p < 0.05, ** p < 0.01.$

Based on Table 7, the direct effect and indirect effect are, respectively, 0.106 and 0.277. It can be seen that after adding the mediating variable of positive emotion to the model with cultural identity as the dependent variable and cognitive evaluation of the natural environment as the independent variable, the impact of cognitive evaluation of the natural environment on cultural identity changed significantly, and the impact of cognitive evaluation of the natural environment and positive emotion on cultural identity remained significant. Therefore, the mediating effect exists, and it is a partial mediating effect. Hypothesis 8 is supported.

As can be seen from Table 8 below: the mediating effect test involves a total of three models, as follows.

Cultural identity (CI) = 1.539 + 0.737 × Cognitive Humanistic Environment Evaluation (CHEE)

Positive emotion = 0.796 + 0.773 × Cognitive Humanistic Environment Evaluation (CHEE)

Cultural identity (CI) = 1.196 + 0.404 × Cognitive Humanistic Environment Evaluation (CHEE) + 0.431 × Positive Emotions (PE)

**Table 8.** The mediating effect test of PE between the CHEE and CI.

| | CI | | | | | PE | | | | | CI | | | | |
|---|---|---|---|---|---|---|---|---|---|---|---|---|---|---|---|
| | B | Standard Error | t | p | β | B | Standard Error | t | p | β | B | Standard Error | t | p | β |
| Constants | 1.593 ** | 0.343 | 4.49 | 0 | - | 0.796 * | 0.396 | 2.009 | 0.045 | - | 1.196 ** | 0.299 | 3.999 | 0 | - |
| CHEE | 0.737 ** | 0.058 | 12.79 | 0 | 0.541 | 0.773 ** | 0.067 | 11.611 | 0 | 0.504 | 0.404 ** | 0.058 | 6.967 | 0 | 0.296 |
| PE | | | | | | | | | | | 0.431 ** | 0.038 | 11.403 | 0 | 0.485 |
| $R^2$ | | 0.293 | | | | | 0.254 | | | | | 0.468 | | | |
| Adjusted $R^2$ | | 0.291 | | | | | 0.253 | | | | | 0.466 | | | |
| F value | | $F_{(1, 395)} = 163.575, p = 0.000$ | | | | | $F_{(1, 395)} = 134.825, p = 0.000$ | | | | | $F_{(2, 394)} = 173.513, p = 0.000$ | | | |

$* p < 0.05, ** p < 0.01.$

According to Table 8, the direct effect and indirect effect are, respectively, 0.404 and 0.333. It can be seen that after adding the mediating variable of positive emotion to the model with cultural identity as the dependent variable and cognitive evaluation of humanistic environment as the independent variable, the effect of humanistic environment cognitive evaluation on cultural identity has changed significantly, and the effect of humanistic environment cognitive evaluation and positive emotion on cultural identity is still significant. Therefore, the mediating effect exists, and it is a partial mediating effect. Hypothesis 9 is supported.

## 5. Conclusions and Implication

### 5.1. Discussion and Conclusions

Based on the theory of positive emotional expansion, this paper constructed a research model of emotional triggers, positive emotional experience, tourists' cultural identity, and heritage conservation behavior. It explored the mechanisms and paths of influence of tourists' emotional experience on cultural identity and tourists' behavior and drew the following conclusions.

Firstly, tourists' cognitive evaluation of the natural and humanistic environment in cultural heritage tourism contexts can induce a positive emotional experience. The results of the empirical study show that tourists' cognitive evaluation of the natural environment, entity caves, and cultural activity experience induce positive emotional experience and have a positive effect. This finding is consistent with the previous theory and studies. The emotional evaluation theory suggests that the process of cognitive evaluation of environmental stimulus is a determinant of the generation of emotions [37]. Zhu et al. [47] and Xu et al. [21] concluded that both a natural and a humanistic environment could stimulate tourists' positive emotions. The empirical results further show that the cognitive evaluation of entity caves has the strongest effect on inducing emotions, followed by that of the natural environment and, finally, the cultural activities experience. Among these, the cognitive evaluation of entity caves directly induces positive negative emotional experience, and the inducing effect is the strongest. In contrast, tourists' cognitive evaluation of the natural environment and cultural experience activity does not induce positive negative emotions. This may be due to the fact that tangible environmental factors such as natural environment and entity caves have a stronger stimulating effect than intangible cultural activities such as cultural activities and play a major role in evoking visitors' emotions [35].

Second, tourists' positive emotional experience positively influences tourists' cultural identity and heritage conservation behavior and plays a key role in the effectiveness of cultural heritage tourism education. In this study, the degree of tourists' positive emotional experience was measured by pleasure, surprise, happiness, and positive negative emotions in a cultural heritage tourism context. The results of the empirical study show that tourists' positive emotions positively influence tourists' cultural identity and heritage conservation behavior and that positive emotions and positive negative emotions have a stronger impact on tourists' cultural heritage conservation behavior. These findings are consistent with the previous studies. Wu et al. [51] took the intangible cultural heritage of local opera as the research object and found that emotional experience based on local opera has a positive impact on cultural identity. Cheng & Chen [29] showed that tourists' emotion to cultural heritage scenic spots has an impact on their responsible environmental behaviors.

Thirdly, positive emotions are part of the mediator variable of the cognitive evaluation of the environment and cultural identity. In the cultural heritage tourism context, positive emotions induced by tourists' cognitive evaluations of the natural and humanistic environment can instantly expand the scope of tourists' attention, perceptions, and actions [33,34,59]. Positive emotions encourage tourists to focus on a wider range of issues, reduce self-attention, increase their awareness of environmentally responsible behavior and build cultural identity [5,6]. The empirical findings suggest that the mediating effect of positive emotions exists in the cultural heritage tourism context and that positive emotional experience plays a mediating role in the cultural identity of visitors and the preservation of cultural heritage.

Fourthly, the cognitive evaluation of the humanistic environment positively impacts cultural identity, which further promotes cultural heritage conservation behavior. The empirical results show that tourists' cognitive evaluation of entity caves and cultural activities positively influences their cultural identity. Compared with the tourists' cognitive evaluation of cultural experience activity, the cognitive evaluation of the entity caves has a stronger effect on the cultural identity, which in turn positively influences their heritage conservation behavior. The entity caves, regarded as the core of Dunhuang Mogao Grottoes, are a comprehensive form of knowledge, technology, art, institutions and beings formed by Chinese cultural history, which act as a crucial bearer of national identity and a crucial testament to its continuation [8]. Through the cultural activity, the social values and collective memory embedded in entity caves are understood, which could enhance cultural and national identity. Therefore, the cognitive evaluation of entity caves and cultural history positively impacts cultural identity. Moreover, Xu et al. [70] have verified through empirical studies that cultural identity is a direct driver of environmentally responsible behavior. This further corroborates the study findings that cultural identity promotes

cultural heritage conservation behavior. In addition, the cognitive evaluation of the entity caves has a stronger effect on cultural identity than the tourists' cognitive evaluation of cultural experience activity, maybe because tangible environmental factors such as entity caves have a stronger stimulating effect than intangible cultural activities such as cultural activities [59].

*5.2. Managerial Implications*

This study has three specific practical implications for managers of cultural heritage tourism attractions and enterprises.

On the one hand, in the cultural heritage tourism context, the positive emotional experience of visitors is influenced by a variety of factors at the destination. The cultural heritage scenic area managers should focus on the creation of the natural environment of the site, as well as the presentation of the unique and authentic humanistic atmosphere of the site, especially of the entity caves. In the presentation of the humanistic atmosphere, it is important to focus on the visibility and interpretation of the heritage itself, but also to develop participatory and integrated cultural experience activity to bring cultural heritage to life in more innovative forms and induce more intense positive emotional experience. By bringing cultural experience "into" the natural environment and increasing the interactivity of cultural experience, the quality of positive emotional experience can be enhanced and tourists' understanding of Chinese culture can be deepened. When developing cultural experience activity in cultural heritage tourism destinations, more innovative means or the interpretation technology should be used to enhance the fun and interaction of cultural experience activity for tourists, especially for teenagers, so that it would be easier to evoke positive emotional experience for tourists.

On the other hand, the different emotional evocations of tourists enhance the education of cultural identity and heritage conservation behavior. The positive emotional experiences of tourists in cultural heritage tourism sites are multidimensional and of varying intensity. Specific emotional dimensions and intensity at various hot and famous sites of a tourist attraction should be measured by scenic area managers, based on the regular browsing routes in the area. Heritage docents or guides should be familiar with the emotional dimensions and intensities of visitors at each of the different popular sites and be actively involved in regulating and stimulating the positive emotional experience of visitors. The positive negative emotions evoked by tourists during cultural heritage traveling, such as grief and regret, are more likely to provoke the tourist to think about his or her behavior and to perceive the positive implications of the negative events compared with positive emotions, such as awe and feeling moved. Cultural heritage tourism managers should pay more attention to the importance of positive negative emotions in educating tourists about cultural identity and heritage conservation behaviors to better fulfill the mission of improving the soft power of national culture.

## 6. Limitations and Future Research

Due to the epidemic, the empirical data in this study were not collected at the scenic spot, and the emotional experience data were collected in the form of tourists' memory, lacking timely emotional feedback from tourists after their visit. Secondly, although a representative cultural heritage tourism site has been selected as a case in the research, the cultural experience activities there are more varied, and different tourists have different choices of cultural activities, so there may be variability in the experience of cultural activities. This paper does not explore the impact of different cultural experience activities on visitors' emotional experience, and there is a lack of consideration as to whether measurements can be used with visitors who are involved in different cultural activities.

To address the limitations mentioned above, subsequent research could focus on the identification of tourists' emotional triggers to a particular cultural experience activity, whether the observable factors such as the form, content, scenic elements, use of technology, interpretation and lively atmosphere of the cultural experience activity have a function of

emotion evoking and the intensity of different factors on the evocation of positive emotions. At the same time, the sample was differentiated using control variables to test whether visitors from different backgrounds and involved in different cultural experiences have different outcomes in terms of their emotional experience. Finally, the evaluation scale of cultural experience activity needs to be further developed to design specific scales for different cultural experience scenarios and to investigate the dimensions of positive emotional experience and the impact on tourists' cultural identity and heritage conservation behavior based on several different cultural experience activities.

**Author Contributions:** Conceptualization, Y.Y., Z.W., H.S. and N.J.; methodology, Z.W and N.J.; software, Z.W.; validation, Y.Y. and N.J.; formal analysis, Y.Y. and Z.W.; investigation, Y.Y. and Z.W.; resources, Z.W. and H.S.; data curation, Y.Y.; writing—original draft preparation, Z.W. and Y.Y.; writing—review and editing, Y.Y. and H.S.; visualization, Y.Y. and N.J.; supervision, H.S.; project administration, H.S.; funding acquisition, H.S. and Y.Y. All authors have read and agreed to the published version of the manuscript.

**Funding:** This research was funded by Social Science Foundation of Shanghai, grant number No. 2020BGL038, National Social Science Foundation of China, grant number No. 19ZD26 and Higher Education Innovation Foundation of Gansu Province, grant number No. 2022B-171.

**Institutional Review Board Statement:** The study was conducted in accordance with the Declaration of Helsinki, and approved by the Institutional Review Board of Fudan University (protocol code FE231731 and date of approval 2023.5.19)." for studies involving humans.

**Informed Consent Statement:** Informed consent was obtained from all subjects involved in the study.

**Data Availability Statement:** The data that support the findings of this study are available from the corresponding author upon reasonable request.

**Conflicts of Interest:** The authors declare no conflict of interest.

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
