# Peer review of "The Impact of Emotional Experience on Tourists’ Cultural Identity and Behavior in the Cultural Heritage Tourism Context: An Empirical Study on Dunhuang Mogao Grottoes"

_sustainability, doi:10.3390/su15118823_

Round 1

Reviewer 1 Report

The study presented offers a correct structure and an in-depth analysis of the objectives set. The theoretical approach is correct and well supported by a correct bibliography. Although it is advisable to review and update the bibliography with the latest work carried out in the field.  Take as an example:

 -Cheng, Zhenfeng, and Xin Chen. "The effect of tourism experience on tourists' environmentally responsible behavior at cultural heritage sites: the mediating role of cultural attachment." Sustainability 14.1 (2022): 565.

 -Peng, Jiamin, et al. "Exploring the influence of tourists' happiness on revisit intention in the context of Traditional Chinese Medicine cultural tourism." Tourism Management 94 (2023): 104647.

 -Câmara, Ester, et al. "Meaningful experiences in tourism: A systematic review of psychological constructs." European Journal of Tourism Research 34 (2023): 3403-3403.

 -Blomstervik, Ingvild H., and Svein Ottar Olsen. "Progress on novelty in tourism: An integration of personality, attitudinal and emotional theoretical foundations." Tourism Management 93 (2022): 104574.

 These are not examples of mandatory compliance, but only a sample of the progress in the field of study proposed. Therefore, an update of the most recent bibliography is required.

 Nevertheless, the state of the art is well presented although it needs to be updated, as mentioned with publications from the last two years.

 Particularly interesting is the case study work, although it would be important to briefly assess some similar published case studies and offer some comparison, perhaps in the area of conclusions.

The methodology, data analysis, results and discussion are correct and yield good and interesting results. Although these results require perhaps a clearer process in the evaluation and improvement of the aspects associated with the site and its visit, possibly in a discussion section and not in the conclusions.

 All in all, this is a good article that requires minor revisions for improvement.

Reviewer 2 Report

Thank you for allowing me to review this manuscript. I find the current manuscript interesting. However, I have some minor suggestions to improve the manuscript:

Abstract

The limitation and suggestion sections should be briefly summarized in a sentence and included in the abstract. The authors should also add R2 scores.

Introduction

The first sentence, p.1 lines 30-33, please rewrite.

P.2 line 115 and line 136 please consider deleting ‘dot’ because there were two dots.

Literature Review

There are no concerns here.

Methods

How did you collect the data?

Which sampling technique did you use and why did you prefer to use this sampling technique?

Did you translate the survey?

Results & Discussion:

Table 6, add R2 scores as a footnote.

Table 7, add direct and indirect effects.

Conclusion:

There are no concerns here.

Overall, a very interesting, well-written, and justified manuscript. All sections, are great and look good. I think it would be a quality addition to Sustainability. I wish the author the best of luck with the revision.
